# Unified Protocol for the Transdiagnostic Treatment of Emotional Disorders in Medical Conditions: A Systematic Review

**DOI:** 10.3390/ijerph18105077

**Published:** 2021-05-11

**Authors:** Jorge Osma, Laura Martínez-García, Alba Quilez-Orden, Óscar Peris-Baquero

**Affiliations:** 1Department of Psychology and Sociology, Universidad de Zaragoza, C/Atarazanas, 4, 44003 Teruel, Spain; 715474@unizar.es (L.M.-G.); 699298@unizar.es (A.Q.-O.); operis@unizar.es (Ó.P.-B.); 2Instituto de Investigación Sanitaria de Aragón, C/San Juan Bosco, 13, 50009 Zaragoza, Spain; 3Mental Health Unit of Tarazona, C/Plaza Joaquín Zamora, 2, 50500 Tarazona, Spain

**Keywords:** Unified Protocol, emotional disorders, medical conditions, health, systematic review

## Abstract

Emotional disorders are those that most commonly present comorbidly with medical conditions. The Unified Protocol for the Transdiagnostic Treatment of Emotional Disorders (UP), a cognitive-behavioral emotion-based intervention, has proven efficacy and versatility. The aim of this systematic review is to know the current (research studies) and future research interest (study protocols) in using the UP for the transdiagnostic treatment of emotional symptoms or disorders (EDs) in people with a medical condition. Using the PRISMA guidelines, a literature search was conducted in Web of Science, PubMed, Medline, and Dialnet. The nine research studies included in this review indicated that the UP is effective in treating emotional symptomatology in a population with a medical condition (effect sizes ranging from *d* = −3.34 to *d* = 2.16). The three included study protocols suggest interest in the future UP application to different medical conditions, and also in distinct application formats. Our review results are encouraging, and conducting more controlled studies is advised to recommend the UP to treat and/or prevent EDs in medical conditions, especially in children and youths.

## 1. Introduction

Communicable diseases are those caused by the transmission of a specific infectious agent from an infected source (person, animal, etc.) to a susceptible host [1], while noncommunicable diseases are the result of a combination of genetic, physiological, environmental, and behavioral factors, and are usually of long duration [2]. Currently, communicable and noncommunicable diseases impose a heavy burden on different countries’ health systems. In 2016, 41 million deaths were due to noncommunicable diseases, which is the equivalent to 71% of deaths per year worldwide. Of this percentage, the majority were due to cardiovascular diseases (17.9 million), followed by cancer (9 million), chronic respiratory diseases (3.9 million), and diabetes (1.6 million) [3]. For communicable diseases, in 2018, 0.8 million deaths were caused by HIV, 1.2 by tuberculosis, and 0.4 by malaria [4].

The comorbidity between communicable/noncommunicable diseases and mental health problems represents another economic burden and a challenge for health systems. It is estimated that the prevalence of mental health problems in this population group is two- to threefold higher than in the healthy population [5]. In relation to these data, a recent meta-analysis reported a 36.6% prevalence of mental disorders in patients with a chronic medical condition [6].

Some studies suggest that the relation between mental health problems and medical conditions is bidirectional [5]. But what does having a comorbid mental health problem with a medical condition imply? Different studies indicate that having a medical and mental condition simultaneously is associated with higher symptom burden (e.g., patients with diabetes are at a higher risk of all-cause mortality, more diabetic complications, and poorer glycemic control), shorter lifetime and worse quality of life and functional impairment, higher rates of health care utilization, and increased medical costs by 45% for each person with a comorbid medical and mental condition [5,7,8]. In turn, comorbidity complicates patients’ help-seeking, course of treatment, and treatment adherence, and thus negatively affects treatment efficacy [9].

The mental disorders that most often present comorbidity with medical conditions are those known as emotional disorders (EDs), a nomenclature that groups mood, anxiety, and related disorders [10]. While the prevalence of EDs varies according to medical condition and severity, ED rates are higher in this population group than in the general population [11]. The results of a study that employed data from the WHO World Health Survey with 245,400 people from 60 countries reported a 23% 1-year depression prevalence for those with one medical condition or two compared to healthy people, whose depression prevalence was estimated at 3.2% [8]. In turn, a study conducted in 17 countries (*n* = 42,249) [12] observed that having a comorbid anxiety disorder and depression was more strongly associated with several medical conditions than both disorders independently. In other words, having a comorbid anxiety and depression disorder increases the risk of having a medical condition at the same time. Many studies have evidenced the presence of EDs that are comorbid with medical conditions, such as cancer [13], type 1 and 2 diabetes [14], chronic pain [15], heart disease [16], obesity [17], and HIV [18], among others.

In light of the above, it is of vital importance to address both comorbid conditions. To date, psychological interventions, particularly those based on cognitive behavioral therapy (CBT), have demonstrated efficacy in improving emotional symptomatology in people with a comorbid medical condition [19,20]. The National Institute for Health and Care Excellence [21] recommends CBT for those patients with a depressive disorder and medical condition at the same time, and the application format varies (group, individual, computerized, or self-applied) depending on depressive symptomatology severity.

When focusing on the efficacy of such interventions, in different meta-analysis studies, CBT has been shown to be effective for treating anxiety and depression in patients with different medical conditions, such as cancer [22], diabetes [23], and Parkinson’s disease [24], among others. Moreover, CBT-based interventions have been shown to improve relevant aspects like quality of life, adherence to medical treatment, coping strategies, and psychosocial adjustment, and to help reduce the use of health services [25,26].

In recent years, CBT has proposed interventions with a transdiagnostic approach that focus on EDs’ shared underlying mechanisms [27]. This group of disorders presents problems in emotional regulation, and shares etiological and maintenance mechanisms like high neuroticism [28]. Several studies suggest that high neuroticism is directly associated with different medical conditions [29,30], which means that it could play a key role in the comorbidity between these and the group of EDs. In addition, high levels of this personality dimension have been associated with increased treatment-seeking in health services for both medical and mental conditions, and could act as a predictor of individuals’ quality of life and longevity [31].

An example of interventions based on this transdiagnostic approach is the Unified Protocol for the Transdiagnostic Treatment of Emotional Disorders (UP), an intervention based on CBT that focuses on treating elevated neuroticism or, in other words, training in adaptive emotional regulation strategies [10]. The UP consists of eight modules, of which five are considered core modules because they address a specific emotional regulation strategy: (1) mindful emotion awareness; (2) cognitive flexibility; (3) opposing emotional behaviors and establishing alternative behaviors; (4) understanding and confronting physical sensations; and (5) emotion exposure. Prior to these, the two initial modules focus on goal setting and motivation enhancement, as well as psycho-education about emotions. Lastly, the final module consists of relapse prevention [10].

By focusing on the common mechanisms in different EDs, the UP offers many advantages over protocols designed to treat specific disorders, including allowing the treatment of people with comorbidity [32] and reducing the costs associated with training mental health professionals in evidence-based psychological treatments (EBPTs) [33]. Moreover, the fact that it is a modular intervention makes it more flexible and adaptable to different problems [34,35] and formats (e.g., individual, group, face-to-face, online) [36,37].

Considering the UP’s efficacy, a meta-analysis and a recent systematic review have shown that UP significantly improves anxious and depressive symptoms after implementation, with effect sizes that are at least comparable to those obtained by disorder-specific interventions [38,39]. They also reported a moderate increase in adaptive emotional-regulation strategies, and a decrease in maladaptive ones. Regarding UP’s efficacy to treat EDs, or anxious and depressive symptoms that are comorbid to medical conditions, we are beginning to find studies with encouraging results [39]. By way of example, a randomized controlled trial (RCT) with irritable bowel syndrome patients reported a significant decrease in not only depressive and anxious symptomatology, but also in postintervention stress levels [40].

The aim of this systematic review is to know the current (publications of conducted research studies) and future (publications of study protocols) research interest in using UP to prevent and treat emotional symptoms or EDs in people also with a medical condition. The obtained results will provide us with relevant information on the medical conditions in which UP has been used or will be used, and also about the characteristics of the conducted research studies and their results. Likewise, the results will allow us to reflect on the obstacles, needs and opportunities that derive from this research line in clinical health psychology.

## 2. Materials and Methods

### 2.1. Search Strategy

The search was done in accordance with the standard set by the Preferred Reporting Items for Systematic Reviews and Meta-Analyses (PRISMA) statement [41] and was preregistered at PROSPERO: CRD42021237138. The search was conducted in February 2021. It included published studies from the electronic databases Web of Science, PubMed, Medline, and Dialnet.

The search strategy included the “Unified Protocol”, “cognitive behavioral therapy”, and “emotional regulation” concepts combined with variations of the terms “physical health” and “medical condition”. Given the diversity of terms, a broad search strategy of terms was used. “Physical health” synonyms were identified and combined with the three concepts with the “AND” Boolean operator. In addition, the “transdiagnostic” term was combined with the “cognitive behavioral therapy” and “emotional regulation” concepts. These terms were searched in titles and abstracts. Appendix B reports the full list of the search terms and combinations used in Web of Science. The same search strategy was used for all the databases (except for Medline and Web of Science, where the search was also included in the subject and keyword fields). The references of the included studies and relevant systematic reviews were searched to identify those studies that were missed during the literature search. The search was conducted in English, except for the Dialnet database, which was conducted in Spanish. There were no language or publication period restrictions.

### 2.2. Inclusion and Exclusion Criteria

The present review included studies that met the following criteria: (1) were scientific articles (including case studies, pilot studies, randomized controlled studies, etc.) or a study protocol; (2) reported using UP or an adaptation of it; (3) included all comparators and outcomes; (4) included samples of all ages with a medical condition and emotional symptomatology or an ED diagnosis. We excluded those studies that: (1) were not scientific articles (e.g., theses, book chapters); (2) did not test a treatment (descriptive studies, systematic reviews, meta-analyses); (3) tested a treatment that was not UP or an adaptation of it; (4) tested UP or an adaptation of it, but not in a sample with a medical condition.

### 2.3. Data Collection

Data collection was performed in two different ways: for research studies and for study protocols. On the one hand, for research studies, data collection was done using a form devised a priori that summarized the following descriptive aspects of each study: author(s), year of publication, country, study design, sample size and characteristics, medical condition, emotional disorders, setting, number and frequency of sessions, primary and secondary measures, and outcomes. On the other hand, the following data were collected from each protocol study using a form devised a priori: author(s), year of publication, country, study design, sample size, medical condition, emotional disorders, format, number and frequency of sessions, primary measures, and secondary measures. Data were collected from all the full texts by one of the authors (L.M.-G.) and discussed with another author (J.O.).

### 2.4. Risk of Bias Assessment

All the studies included in this review were independently rated for quality by two reviewers (L.M.-G. and A.Q.-O.), except for those which were study protocols. If the rating differed, reviewers discussed the articles to reach a consensus with a third reviewer (J.O.). The Study Quality Assessment Tools from the National Heart Lung and Blood Institute [42] were employed to assess study quality and the risk of bias. This set of tools allows reviewers to rate studies as “good”, “fair”, or “poor”. It was preferred because it includes six types of studies and specific criteria according to the study design (i.e., Controlled Intervention Studies, Systematic Reviews and Meta-Analyses, Observational Cohort and Cross-Sectional Studies, Case–Control Studies, Before–After Studies With No Control Group, and Case Series Studies). The total quality scores ranged from 9 to 14 points, depending on the study design.

## 3. Results

### 3.1. Search and Screening

Initially, 1507 publications were identified from the database searches and after screening reference lists. As Figure 1 illustrates, after excluding duplicates (*n* = 607), 900 publications remained for screening. After the initial screening of titles and abstracts, 865 of these documents were excluded based on the inclusion and exclusion criteria. Full texts were reviewed for the remaining 35 publications. After an eligibility assessment made up the full texts, 23 publications were excluded, which gave a final sample comprising 12 publications, nine of which were research studies and three were study protocols.

The search, screening, and data collection processes were conducted independently by two of the authors (L.M.-G. and A.Q.-O.). Previously, four questions were determined, one per exclusion criteria: (1) Is it a scientific article? (2) Does it test a treatment? (3) Does it apply the UP? (4) Does it apply the UP in a sample with a medical condition? The excluded publications were grouped according to all four exclusion reasons (see Figure 1). If in doubt, study eligibility was discussed with another author (J.O.). After the study eligibility assessment phase, inter-rater agreement was calculated (Cohen’s kappa). A 100% agreement (Cohen’s Kappa = 1) was reached.

### 3.2. Characteristics of the Included Research Studies

The characteristics of the included research studies are shown in Table 1. Of the nine research studies included in the systematic review, only one was a prevention study [43], while the rest were treatment studies. Most were published in the USA (*n* = 4) [43,44,45,46] and Iran (*n* = 4) [40,47,48,49], while the remaining study was published in Sweden [50], between 2012 and 2020. The sample size of the different research works ranged from 2 to 70 participants. Regarding the sample’s characteristics, in the vast majority of studies (*n* = 8), the participants were adults aged between 20 and 79 years. Only in one of the included articles were the participants aged under 18 years [46]. In relation to the sample’s genders, three studies included only women [43,47,48] and one included an exclusively male sample [45]. Of the remaining five [40,44,46,49,50], the proportion of participants was mostly women, 60% versus 40%.

The included research studies targeted different medical conditions, specifically: breast cancer [46]; irritable bowel syndrome [50]; HIV [44]; chronic pain [44,50]; idiopathic Parkinson’s disease [46]; multiple sclerosis [48,49]; and infertility problems [47]. The most frequent EDs were anxiety disorders and depression. More specifically, the participants in the nine studies had symptomatology or had been diagnosed with generalized anxiety, social anxiety, agoraphobia, panic disorder, dysthymia, or major depressive disorder.

In design terms, two were pilot studies done with a single group [43,45], two were a single-case experimental design [46,50], one was a case study [44], and four were RCTs [40,47,48,49]. In the RCTs, the control condition was a waiting list in one [40], in which they received usual pharmacological treatment, as well as information about the medical condition, and an active condition in the remaining three research studies. The three active control conditions were a mindfulness-based psychological intervention (MBSR) [47] and two psychosocial interventions: psycho-educational and supportive [48,49]. In turn, five designs included follow-up measures at 3 months [44,45,47,48,50] and one study conducted follow-up at 6 weeks after the intervention ended [46]. The remaining three research studies collected pre- and post-treatment measures [40,43,48], and one also collected measures during the intervention before each session [43].

Most research studies were conducted in outpatient settings, such as hospital and university clinics. The intervention was delivered face-to-face in six of the research studies [40,44,45,47,48,49]. In the other three, one combined face-to-face (50%) and telephone (50%) sessions [43], while another gave the option of receiving the intervention face-to-face or online via Webcam [46]. In the remaining research study, the intervention was self-applied over an Internet platform and was accompanied both by telephone calls for support/clarifying doubts at the beginning of each module and by e-mail feedback [50]. The intervention format was individual in six of the studies [40,43,44,45,46,50] and group-based in the remaining three [47,48,49]. Intervention intensity (i.e., frequency) differed among research studies. Interventions ranged from four sessions in the least intensive to 17 in the most intensive. Periodicity was weekly in all the research studies, except for one with 2-week intervals between sessions [43]. Finally, session duration ranged from 50 min per session to 2 h. In those research studies in which the intervention was performed by telephone or accompanied by telephone calls, they lasted from 15 min to 1 h.

### 3.3. Modifications to UP

While four of the nine research studies included in this systematic review reported having applied the original UP [10], adaptations were made in the other five studies to both content and the number of sessions or intervention duration. The content was adapted to the context of the participants’ medical condition in three research studies. One applied the “Unified Protocol for the Treatment of Emotions in Youth with Pain (UP-YP)” to adolescents with chronic pain [44], another used the ESTEEM-SC in an HIV population [45], and the remaining applied the “Unified Protocol for Prevention of Depression after Cancer (UP-PDAC)” to women with breast cancer [43]. The latest adaptation of UP significantly reduces the number of sessions that are normally held throughout the protocol, between 12 and 16 [10], by leaving four sessions. In relation to the structure of sessions, in the research study with patients with Parkinson’s disease, an optional session for each participant’s family member or partner was included, with psycho-educational content on the emotions and skills trained by the UP [46]. Finally, in one research study that included people with chronic pain, where the intervention format was self-applied via the Internet, the UP patient workbook content (psycho-educational texts and number of exercises) was reduced to create a workbook in which psycho-educational content and examples related to chronic pain were added [50].

### 3.4. Clinical Effectiveness in Emotional Disorders and/or Emotional Symptomatology

As shown in Table 1, most research studies reported that UP proved effective in significantly reducing emotional symptomatology severity; i.e., anxious and depressive symptomatology, in people with a medical condition comorbid with an ED or emotional symptomatology. Effect sizes ranged from *d* = −3.34 to *d* = 2.16. One research study also observed that these results were obtained with both face-to-face and online UP applications [46]. Some research studies report changes in emotional regulation, with effect sizes ranging from *d* = 0.44 to *d* = 1.40. In one of them, improvements in the specific strategies of cognitive reappraisal (*d* = 1.32) and emotional suppression (*d* = 1.04) were observed [40].

As indicated in previous paragraphs, three of the four included RCTs compared UP to an active control condition. In two with a multiple sclerosis population, anxious and depressive symptomatology, and emotional dysregulation and tendency to worry, they significantly improved in those who received the UP compared to a psycho-educational and supportive intervention [48,49]. In turn, in those assigned to the UP condition, both these studies reported a smaller negative affect, with effect sizes of *d* = −2.21 [48] and *d* = 1.89 [49], and an increasing positive affect with effect sizes of *d* = 1.46 [48] and *d* = 1.51 [49]. Both were significant. In the third research study performed with women with infertility problems, the UP appeared to be equally as effective in decreasing anxious and depressive symptomatology as the MBSR program. However, the reported results did not indicate whether there was any significant difference between both conditions [47].

### 3.5. Clinical Effectiveness in Medical Symptoms

As shown in Table 1, only three research studies reported changes in the medical condition measures presented by the participants, two with a chronic pain population and one with people with irritable bowel syndrome. For chronic pain, in one research study pain intensity decreased in the two cases it included [44]. However, this decreased pain was minor and occurred at follow-up in both cases, with increased pain at the post-intervention in one of the cases and remained in the other. The pain results after applying an adaptation of UP over the Internet indicated that pain intensity remained the same in some participants, but increased in others at post-intervention and at 3-months follow-up [50]. In the population with irritable bowel syndrome, the UP seemed effective in improving gastrointestinal symptoms. It indicated large effect sizes compared to the control group [*d* = 1.33 (0.63–1.69), 95% CI]. This improvement seemed mediated by changes in emotional regulation, specifically improvements in cognitive reappraisal strategy and decreased emotional suppression [40].

### 3.6. Patient’s Opinion after the UP Intervention

Regarding participants’ opinion assessment after the intervention, only two of the nine research studies collected this information. The first included acceptability and satisfaction with treatment, with high levels (*M* = 4.6/5 and 30.9/32, respectively), as well as suggestions to improve the intervention. The collected answers were “greater flexibility regarding number of treatment sessions” or “additional modules on motivation or increased caregiver involvement” [46]. The second identified the barriers that participants found during the intervention. The collected answers were “logistical issues related to scheduling” or “therapist characteristics”, which could be addressed by providing flexible scheduling or different intervention formats (i.e., phone or online), and by offering different choice of therapist possibilities [45].

### 3.7. Risk of Bias Assessment

As observed in Appendix C, the research studies included in this review could be classified into three of the categories proposed by the NHLBI (case studies, before–after studies with no control group, or controlled clinical trials) [38]. The quality of the research study classified as a case study [44] was assessed as “fair”, with a score of 6 out of a maximum of 9 points. This rating was because the study neither describes the statistical methods, nor reports whether cases were consecutive, and only a 3-month follow-up was performed. Second, of the four before–after studies, two were rated as “good” quality research [43,46], as both scored 9 points out of a maximum of 12; the quality of the remaining two [45,50] was rated as “fair”, with 7 and 8 points. Neither article reported whether the sample size was large enough to provide reliable findings, or if the people evaluating the results were blinded to participants’ interventions. Finally, one of the controlled intervention studies [47] was rated as “poor” quality. Despite being described as an RCT, it did not follow up most criteria for controlled studies (i.e., randomization method, allocation, blinded assessment) or aspects such as dropout rates, adherence to treatment, or participants’ demographic characteristics. The quality of the remaining three research studies [40,48,49] was “good”, with scores of 13, 12, and 13 points out of a maximum of 14.

### 3.8. Future Directions

Table 2 shows the characteristics of the included study protocols. In response to the question about what the future research interest is in the UP application line in medical conditions, the three study protocols identified in the above-described search were analyzed. First, each protocol was developed in a different country. One, published in 2016, describes a study to be conducted in Australia [51]. The remaining two, published in 2020, describe studies to be performed in Spain [52] and the USA [53].

Each study protocol focuses on applying the UP in adults with emotional symptomatology or EDs, and different medical conditions; specifically, in people with obesity who undergo bariatric surgery [52], urinary problems [53], and cardiovascular diseases [51], respectively. The intended sample size for each study will range from 40 to 200 participants, who will be recruited in all the three cases in a hospital setting. The sample will comprise both genders in two studies [51,52] and will be limited to only women in the remaining study [53].

Regarding the study design included in protocols, two will be RCTs [51,53] and the third will be a single-case experimental design [52]. All three protocols include a chronology of the assessments to be made throughout studies. All include baseline, in-treatment, post-treatment, and follow-up measures. Regarding the latter, one of the studies will conduct follow-up assessments every 3 months up until 2 years from the end of the intervention [52], while the remaining two include follow-up at 6 months [51], and at 3 and 6 months [53], respectively. Regarding the control conditions in RCTs, in one UP will be compared to a supportive therapy program [53], while the control group will receive an education pack in the other [51]. In the latter, the design includes, in addition to UP and the control condition, a nonrandomized comparison cohort in which those participants who do not present anxious and/or depressive symptomatology, and therefore do not meet all the inclusion criteria to be randomized to one of the first two conditions, will be included.

The UP application format described in each protocol is individual in one [53] and group in another [52]. The remaining protocol does not report this aspect [51]. Intervention will be applied face-to-face in two studies [51,53], while that which will implement the UP in the group format will do so online on the Cisco Webex platform [52]. The two studies that will implement the face-to-face intervention will do so in a hospital outpatient setting [51] and on a university campus [53]. UP will be applied weekly during 12 sessions in three studies, although one of them clarifies that between 12 and 18 sessions will be held, depending on the number of sessions spent on each core module [51]. Finally, sessions will last between 45 min [53] in one study and 2 h in another [52]. The third of the protocols does not report this aspect [51].

## 4. Discussion

The aim of this systematic review is to know the current and future research interest in using the UP for preventing and treating emotional symptoms or EDs in people with a medical condition. To date, although two systematic reviews, one with a meta-analysis, have collected existing evidence for the effectiveness of the UP applied to an adult population [38,39], this is the first systematic review that focuses exclusively on applying the UP to people with a medical condition in addition to emotional symptomatology or ED.

The results of this study indicate that, to date, the UP has been used in populations with seven different medical conditions (chronic pain, infertility, irritable bowel syndrome, multiple sclerosis, HIV, idiopathic Parkinson’s disease, and breast cancer). Based on registered study protocols, it is planned to be used in three more medical conditions (obesity, urinary problems, and heart disease). Only 12 studies met the inclusion criteria, which suggests that applying the UP in clinical health psychology requires much more research. The overall results show that the UP is effective in improving emotional symptomatology in most studies, as well as medical symptomatology in some health conditions in, for example, gastrointestinal symptoms [40]. In this study, the conducted mediation analysis suggests that targeting emotional regulation may be beneficial for patients with comorbid emotional and medical symptomatology by mediating emotional regulation in improvements in both symptomatology types [40]. This result falls in line with those works that indicate the bidirectional relationship between emotional and physical symptoms [5]. However, these data are preliminary and cannot be generalized, and much more research in this line is needed with quality studies and larger samples to clarify the findings and to accumulate evidence on this matter.

One aspect to note is that, to date, the intervention with children and adolescent populations with medical problems has drawn less interest from researchers. If we consider, on the one hand, that anxious and depressive symptomatology levels are high in children and adolescents with chronic medical illnesses [54,55] and, on the other hand, only one of the included studies applied the UP to the adolescent population [44], and none did to the pediatric population, we underline the need to conduct studies that investigate the UP efficacy in this population, especially when none of the three study protocols foresees applying it to these age groups.

Four of the nine included research studies were categorized as fair risk of bias because the quality of three was rated as “fair” [44,46,50] and that of the remaining fourth as “poor” [47], with a high bias risk in the latter. In addition, most included research studies were pilot or case studies, while only four were RCTs. The information from RCTs is equally valuable, and their decision to conduct such research is understood, as it is a novel and scarcely studied research line, so more RCTs in this field are necessary because their internal validity is higher. Thus, the results of the review done of study protocols are encouraging because we observed that future research interest lies in this direction, as two of the three reports are about conducting RCTs [51,53].

Another aspect worth mentioning is that treatment studies have drawn more interest than prevention studies. In addition, all the research studies included in this systematic review applied the UP in private or university clinics, and none was performed in naturalistic settings like public health services. This is an obstacle to disseminate EBPTs to all the people in need and goes against the United Nations’ 2030 agenda recommendations, specifically Objective 3: “Health and Well-Being”, which proposes universal access to public health care [56], and the recommendations of the World Health Organization (WHO) on access to evidence-based interventions [57]. So, it can be stated that some future interest lies this direction, as one included study protocol is committed to implement it in this setting [52]. However, we encourage researchers to conduct studies that apply the UP in public hospitals and clinics to evaluate its implementation and efficacy in these naturalistic settings.

Furthermore, only two of the nine research studies included information about patients’ opinions of received treatment. It is necessary to know patients’ opinions about the interventions they receive to ensure correct implementation [58] and, given the high degree of acceptability, it is directly related to intervention effectiveness [59]. In addition, five of the included research studies made modifications to the original UP, which demonstrates the excellent versatility of this intervention. We found different modifications made to the UP in the research studies, where its ability to adapt to the specific characteristics of each medical condition and the needs of those people who suffer from them are noteworthy. For example, some research studies incorporate content related to the disease or related aspects (e.g., minority stress model) [45] or reduce the number of sessions to be intensively applied [43]. Thus, the flexibility and adaptability characteristics of the UP are factors that facilitate EBPT dissemination.

Another interesting aspect to mention is that only three research studies applied the UP as group intervention. Their results indicated that this format facilitates the normalization of medical condition experiences and reduces the stigma associated with EDs and their treatment [49]. This falls in line with previous studies that have indicated that, by identifying other group members, the sharing of experiences and normalization of experiences is facilitated [60]. In addition, the application of the UP in a group format has benefits for public health systems by improving cost-benefit relations and cutting waiting lists. This allows clinicians to attend several people at the same time [61]. Therefore, as individuals with a medical condition plus ED seem to benefit from this application format, future research along these lines is needed to provide more data about effectiveness and distinct advantages compared to individual UP application. Future interest in group UP applications with this population comes over because one study included protocols that contemplate this format [52].

Similarly, only two research studies and one included protocol propose applying the UP via the Internet, and both include online webcam sessions and a self-applied format with phone calls. The results of one of the research studies show that these do not differ from those obtained by in-person UP applications [46], which falls in line with previous findings indicating that treatments guided by the Internet can be as effective as face-to-face treatments [62]. As previously mentioned, one of the study protocols proposes an online UP application, which denotes future interest in the scientific community applying this format which, in addition to offering the advantage of providing more treatment accessibility, is more recommendable and safer in the today’s COVID-19 pandemic [63].

Finally, some limitations should be taken into account when interpreting the results of this systematic review. First, a limited number of studies have been reported. In addition, most research studies were pilot or case studies, as only four were RCTs, but they were the most rigorous and internally valid. Furthermore, not all the included research studies were rated after being evaluated with the NHLBI tool as being of “good” quality, hence a bias risk is posed. In turn, some factors may have biased the present systematic review findings, including the fact that only four databases were used for searches, and given the possibility of some studies offering contradictory results to those herein presented, which could change our conclusions. Finally, it is important to note that this systematic review is limited to the authors’ interpretations.

Despite the above-discussed limitations, the results of this systematic review have important clinical implications. The reviewed studies generally indicate that the UP is effective for treating emotional symptomatology in a population also with a medical condition, which implies improvements in medical symptomatology in some research studies. Although the results herein presented are preliminary and require further replication, these, together with UP’s flexibility and its ability to adapt to different existing medical conditions, indicate that the UP can be a useful psychological intervention with this population group.

## 5. Conclusions

In conclusion, the present systematic review provides preliminary data about UP’s feasibility and clinical utility for the transdiagnostic treatment of emotional symptomatology or EDs in people also with a diagnosed medical condition. Despite the encouraging results, we identified some gaps that research should address in future studies, such as applying the UP to more medical conditions, children, and young samples, in cost-effective formats (group and online) and with more sophisticated research designs. In addition, more prevention studies are needed. Clinical health psychologists, especially those working in public health settings, can benefit from this transdiagnostic proposal thanks to its cost-effectiveness and versatility.

## Figures and Tables

**Figure 1 ijerph-18-05077-f001:**
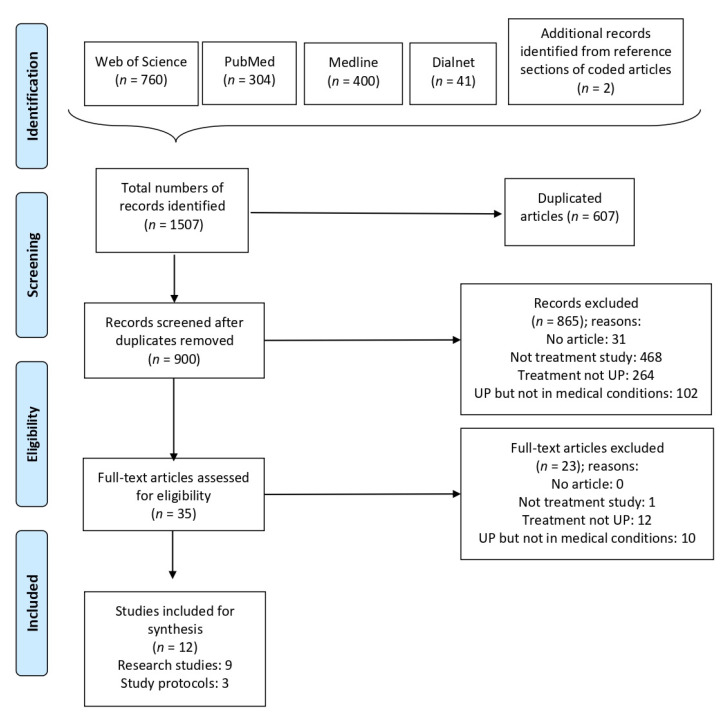
Flow diagram of the study selection following the PRISMA guidelines (Appendix A) [41].

**Table 1 ijerph-18-05077-t001:** Characteristics of the included research studies.

Reference, Country and Publication Year	Sample (*N*)	Study Design and Setting	Medical Condition	Emotional Disorder	Length/Session Frequency	Primary and Secondary Measures	Results
[44] USA, 2012	2 Tx Age: 14, 17 Gender: 1 woman and 1 man	Case study/ Individual—Outpatient	Chronic pain	SAD, MDD	12 and 17 sessions/50 min/1 per week	CSI: somatization; EESC: emotional awareness and expression; FPS-R: pain intensity; FDI: functional disability; RCADS (total anxiety and depression, social anxiety and depression subscales).	Case 1: improvements pre-to-post in FDI, RCADS (total and social anxiety), EESC, CSI and pain level maintained (FPS-R = 8). Case 2: slight improvement in RCADS (total) at post-, with no changes in RCADS (depression) and worse EESC, CSI and FPS-R levels. At the 3-month follow-up, significant improvements in all the levels were observed in both cases (RCADStotal from 31 to 17 and from 44 to 37; EESCce from 18 to 14 and 22 to 14; EESCee from 22 to 18 and from 36 to 28; CSI from 12 to 9 and from 32 to 18; FPS-R from 8 to 6 and from 6 to 2).
[47] Iran, 2019	45 (15 Tx1. + 15 Tx2 + 15 Cont.) Age: 20–45 Gender: 100% woman	RCT/Group—Outpatient	Infertility	Anxious and depressive symp.	UP (Tx): 10 sessions/2 h/1 per week MBSR (Cont.): 8 sessions/2 h/1 per week	BAI: anxious symptoms; BDI-II: depressive symptoms; DERS: emotional dysregulation; IUS-12: intolerance to uncertainty.	Improved anxious and depressive symp. at post- and 3-month follow-up for the UP condition and in MBSR. Reported results do not explain if there was any significant difference between both conditions. Significantly better improvements in the UP group compared to the waiting list condition (anxious and depressive symp. increased at post- and 3-month follow-up in the latter).
[40] Iran, 2018	64 (32 cont. + 32 Tx) Age: 30.9 Gender: 59.38% woman 40.63% man	RCT/Individual—Outpatient	Irritable bowel syndrome	Anxious and depressive symp.	12 sessions/2 h/1 per week	DASS-42: anxious, depressive and stress symptoms; ERQ: emotional regulation strategies; GSRS: severity of intestinal symp.	Significant decrease in depression, anxiety, stress, gastrointestinal symp., and significant improvement in cognitive reappraisal and emotional suppression strategies with a marked effect sizes: between-group comparison (Cohen’s *d* between 0.97 and 1.34) and under the UP condition (Cohen’s *d* between 0.92 and 1.18). No significant differences found for the waiting list condition from pre- to post-treatment for any measure. Mediation analyses indicated that changes in emotional regulation mediated the effect of UP on emotional and gastrointestinal symptomatology.
[48] Iran, 2020	64 (32 Cont + 32 Tx) Age: 35.13 Gender: 100% woman	RCT/Group- Outpatient	Multiple sclerosis	Anxiety and/or depressive disorder	UP: 14 sessions/2 h/1 per week TAU: 14 sessions/2 h/1 per week	DERS: emotional dysregulation; HADS: anxious and depressive symptoms; PANAS: positive and negative affect; PSWQ: tendency to worry.	Significant improvement for the UP condition of depressive symp., anxious symp., tendency to worry, emotional dysregulation, and positive and negative affect compared to the control group (Cohen’s *d* ranged from 0.45 to 2.34).
[49] Iran, 2020	70 (35 Cont. + 35 Tx.) Age: 35.30 Gender: 61.43% woman 38.57% man	RCT/Group—Outpatient	Multiple sclerosis	GAD, SAD, MDD and PDD	UP: 12 sessions/2 h/1 per week TAU: 12 sessions/2 h/1 per week	DERS: emotional dysregulation; HADS: anxious and depressive symptoms; PANAS: positive and negative affect; PSWQ: tendency to worry.	At post-treatment, significant improvement in the UP condition of depressive symp, anxious symp., positive and negative affect, emotional dysregulation, and tendency to worry compared to the control condition (Cohen’s *d* ranged from 0.44 to 2.16). Improvements remained at the 3-month follow-up.
[45] USA, 2017	11 Tx Age: 34.4 Gender: 100% man	Single-condition pilot study/Individual—Outpatient	HIV	Anxious and depressive symp.	10 sessions/1 per week	CES-D: depressive symptoms; DERS: emotional dysregulation; OASIS: anxious symptoms; ODSIS: depressive symptoms; PSWQ: GAD symptoms; SCS: sexual compulsivity; SIP-DU: frequency of negative consequences of drug use; TLFB: sexual behavior and substance use; YBOCS: obsessive-compulsive symptoms.	Significant reduction with large effect sizes of anxious, depressive, and obsessive-compulsive symptomatology from the baseline to the 3-month follow-up (Cohen’s *d* ranged from 0.83 to 1.68). Reduced medium effect sizes, albeit not significant, in emotional dysregulation, functional impairment in relation to anxiety, and number of male sexual partners and condomless anal sex (Cohen’s *d* ranged from 0.49 to 0.61). Reduction in small to medium, but not significant, effect sizes of sexual compulsivity, functional impairment in relation to depression, drug use, and problems with drug use (Cohen’s *d* ranged from 0.21 to 0.41).
[46] USA, 2019	9 Tx Age: 61.22 Sex: 55.56% woman 44.44% man	SCED/Individual—Outpatient or the Internet	Idiopathic Parkinson’s disease	AG, GAD, SAD, MDD, PD and PDD	12 sessions/50–60 min/1 per week	ADIS-V: ED; AS: apathy; BAI: anxious symp.; BDI-II: depressive symp.; CSQ-8: satisfaction with treatment; Telepresence on Videoconference Scale; FES: self-efficacy and worry about falling; GDS: depressive symp.; OASIS: anxious symp.; ODSIS: depressive symp.; STAI: anxious symp.	Statistically significant decrease in anxious and depressive symp. in 7 of the 9 participants at the post- and 6-week follow-up. Significant reduction in fear of falling in two participants (with high scores in the pre-) at post- and follow-up. Significant reduction in apathy in two participants at post- and follow-up, and increase in one participant. High satisfaction with treatment (M CSQ-8 = 30.9). The results did not differ for session modality (online or face-to-face).
[43] USA, 2019	15 Tx Age: 57 Gender: 100% woman	Single-condition pilot study/Individual—Outpatient and telephone	Breast cancer	Depressive symp.	4 sessions/2 h (face-to-face) 45 min (telephone)/1 per week with 2 weeks between sessions	ACS: fear of depression; CES-D: depressive symp.; COPE ACCEPTANCE: cancer-related acceptance; COPE AVOID: cancer-related avoidance; DTS: discomfort tolerance; EAC: emotional expression; FFMQ: description of emotions and thoughts and nonjudgment; MEAQ: experiential avoidance; RRQ: rumination; UP CSQ: cognitive skills.	Large effect size on cancer-related acceptance strategy (Cohen’s *d* = 0.82); medium effect on cancer-related emotional expression (Cohen’s *d* = 0.65) and smaller effects on cancer-related avoidance (Cohen’s *d* = 0.32) and depressive symp. (Cohen’s *d* = 0.42).
[50] Sweden, 2017	5 Tx Age: 46.40 Gender: 60% woman 40% man	SCED/Individual—Internet + telephone	Chronic pain	AG, GAD, SAD and MDD	10 sessions/self-applied; approx. 1 module per week	MINI: DSM-V diagnosis; OASIS: anxious symp.; ODSIS: depressive symp.; ÖMPSQ-sv: pain intensity and coping problems. Satisfaction with treatment, treatment completion and compliance, and self-report improvement in strategies trained by PU.	Improvements in anxious and/or depressive symp., in four of the five participants, with medium to large effects, but only significant in two participants. At post-, P3 and P5 continued to meet the criteria for the same ED diagnosis, but P1 and P4 no longer met the criteria for any ED. In pain intensity, increases or no change at post- and 3-month follow up. High satisfaction with treatment. Patients reported improvements in each PU strategy.

Note: ACS, Affective Control Scale; ADIS-V, Anxiety Disorders Interview Schedule; AG, Agoraphobia; AS, Apathy Scale; BAI, Beck Anxiety Inventory; BDI-II, Beck Depression Inventory; Charact., Characteristics; CES-D, Center for Epidemiological Studies Depression Scale; Cont., Control Group; COPE Inventory, Cancer-related Acceptance and Avoidance Subscales; CSI, Children’s Somatization Inventory; CSQ-8, Client Satisfaction Questionnaire; DASS-42, Depression, Anxiety, and Stress Scale; DERS, Difficulties in Emotion Regulation Scale; DTS, Distress Tolerance Scale; EAC, Emotion Approach Coping: Emotion Expression Subscale; ED, Emotional Disorder; EESC, Emotion Expression Scale for Children; ERQ, Emotion Regulation Questionnaire; FDI, Functional Disability Inventory; FES, Falls Self-Efficacy Scale; FFMQ, Five-Factor Mindfulness Questionnaire: Describe and Nonjudging Subscales; FPS-R, Faces Pain Scale—Revised; GAD, Generalized Anxiety Disorder; GDS, Geriatric Depression Scale; GSRS, Gastrointestinal Symptoms Rating Scale; HADS, Hospital Anxiety and Depression Scale; IUS-12, Inventory Intolerance of Uncertainty Scale—Short Form; MBSR, Mindfulness-Based Stress Reduction; MDD, Major Depressive Disorder; MEAQ, Multidimensional Experiential Avoidance: Avoidance and Repression Subscales; MINI, Mini International Neuropsychiatric Interview; NAP, Nonoverlap of All Pairs; OASIS, Overall Anxiety Severity and Impairment Scale; ODSIS, Overall Depression Severity and Impairment Scale; ÖMPSQ-sv, Örebro Musculoskeletal Pain Screening Questionnaire; PANAS, Positive and Negative Affect Schedule; PD, Panic Disorder; PDD, Persistent Depressive Disorder; PSWQ, Penn State Worry Questionnaire; RCADS, Revised Child Anxiety and Depression Scale; RCT, Randomized Controlled Trial; RRQ, Rumination Subscale of the Rumination and Reflection Questionnaire; SAD, Social Anxiety Disorder; SCED, Single-Case Experimental Design; SCS, Sexual Compulsivity Scale; SIP-DU, Short Inventory of Problems for Drug Use; STAI, State–Trait Anxiety Inventory; Symp., Symptomatology; TAU, Treatment As Usual; TLFB, 90-Day Time Line Follow Back; Tx, Treatment Group; UP, Unified Protocol for Transdiagnostic Treatment of Emotional Disorders; UP CSQ, Unified Protocol Cognitive Skills Questionnaire; YBOCS, Yale–Brown Obsessive Compulsive Scale.

**Table 2 ijerph-18-05077-t002:** Characteristics of the included study protocols.

Reference, Country and Publication Year	Sample (*N*)	Study Design and Setting	Medical Condition	Emotional Disorder	Length/Session Frequency	Primary Measures	Secondary Measures
[52] Spain, 2020	60	SCED/Group—Internet	Obesity	Anxious and/or depressive symp. or at least a diagnosis of ED	UP: 12 sessions/2 h/1 per week	BAI: anxious symp.; BDI-II: depressive symp.; BMI: weight gain or loss; MINI: primary and secondary diagnosis of ED.	PANAS: positive and negative affect; NEO-FFI: extraversion and neuroticism; QLI: quality of life; EuroQol: health-related quality of life; MI: negative impact of health problem and ED in areas of daily life; DERS: emotional dysregulation; BEAQ: experiential avoidance; PHLMS: present awareness and acceptance; ERQ: cognitive reappraisal and expressive suppression; BITE: bulimia symp. and signs and symp. associated with binge eating; BSQ: fear of gaining weight, low self-esteem in relation to appearance, desire to lose weight, and body dissatisfaction; EES: tendency to use food to cope with negative affect; STQ: satisfaction with treatment.
[53] USA, 2020	40 (20 Tx + 20 Cont.)	RCT/Individual—outpatient	Urinary problems	Anxious symp. or Anxiety disorder	UP (Tx): 12 sessions/45 min./1 x week Supportive ther.: 12 sessions/45 min/1 x week	PROMIS-29 (Anxiety subscale); UDI-6: urinary problems.	Mini-IPIP: big 5 personality traits; PCL-5: trauma history; PGI-I: improvement of urinary symp.; PROMIS-29 (depression, fatigue, pain, physical functioning, sleep disturbances, social roles subscales); RRS: ruminative style.
[51] Australia, 2016	200 (25 Tx + 25 Cont. + 150 Comparative cohort)	RCT/lndividual—outpatient	Cardiovascular diseases	AG, GAD, SAD, MDD, PDD, PTSD, PD	UP (Tx): 12–18 sessions/1 per week EUC (Cont.): Educational package	GAD-7: GAD symp.; OASIS: anxious symp.; PHQ-9: depressive symp.; and SF-12: quality of life.	AUDIT-C: alcohol use; DASS-21: stress; GATS: tobacco use, CVE; MINI: ED diagnosis; MOS SAS: adherence to treatment.

Note: AG, Agoraphobia; AUDIT-C, Alcohol Use Disorders Identification Test-Shortened Clinical Version; BAI, Beck Anxiety Inventory; BDI-II, Beck Depression Inventory; BEAQ, Brief Experiential Avoidance Questionnaire; BITE, Bulimic Investigatory Test; BSQ, Body Satisfaction Questionnaire; Cont., Control Group; DASS-21, Depression, Anxiety, and Stress Scales; DERS, Difficulties in Emotion Regulation Scale; CVE, Cardiovascular Event; ED, Emotional Disorder; EES, Emotional Eating Scale; ERQ, Emotion Regulation Questionnaire; EUC, Enhanced Usual Care; EuroQol, European Quality of Life Scale; GAD, Generalized Anxiety Disorder; GAD-7, Generalized Anxiety Disorder Scale; GATS, Global Adult Tobacco Survey; BMI, Body Mass Index; MDD, Major Depressive Disorder; MI, Maladjustment Inventory; MINI, Mini-International Neuropsychiatric Interview; Mini-IPIP, Mini-International Personality Item Pool; MOS SAS, Medical Outcomes Study Specific Adherence Scale; NEO-FFI, NEO Five-Factor Inventory; OASIS, Overall Anxiety Severity and Impairment Scale; PANAS, Positive and Negative Affect Schedule; PCL-5, PTSD Checklist for DSM-5 with Life Events Checklist; PD, Panic Disorder; PDD, Persistent Depressive Disorder; PGI-I, Patient Global Impression of Improvement; PHLMS, Philadelphia Mindfulness Scale; PHQ-9, Patient Health Questionnaire; PROMIS-29, Patient-Reported Outcomes Measurement Information System; PTSD, Post-Traumatic Stress Disorder; QLI, Quality of Life Index; RCT, Randomized Controlled Trial; RRS, Ruminative Responses Scale; SAD, Social Anxiety Disorder; SCED, Single-Case Experimental Design; SF-12, Short-Form Health Survey; STQ, Satisfaction with Treatment Questionnaire; Symp., Symptomatology; Ther., Therapy; Tx, Treatment Group; UDI-6, Urinary Distress Inventory; UP, Unified Protocol for Transdiagnostic Treatment of Emotional Disorders.

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
