# Peer review of "Unified Protocol for the Transdiagnostic Treatment of Emotional Disorders in Medical Conditions: A Systematic Review"

_ijerph, 2021, doi:10.3390/ijerph18105077_

Round 1

Reviewer 1 Report

The authors are commended for this interesting study. Here are my comments and questions:

  1. This systematic review attempts to collate all empirical evidence from the 9 studies that fit its specified eligibility criteria.  This reviewer thinks the number of studies in this systematic review is too few to address a systematic review research objective.  
  2. The nine studies include RCT (four studies), PILOT Studies (two), a single-case experimental study (two) and a case study(one), three study protocols,  and the authors end mixing up apples and oranges in their analysis and synthesis.
  3. Case studies can be useful in research but they are at the very bottom of an evidence-based conclusion and pilot studies are typically used to gather evidence to refine a planned study design, not to use the study findings as evidence basis. Study protocols do not constitute an evidence base. Thus if we eliminate the case studies, study protocols, and pilot studies, the number of studies is even fewer.
  4. The systematic review should look into all the published articles across many years within the year limit of publication set in the eligibility criteria. What was the year limit of publication set in the eligibility criteria?
  5. Page 12, lines 398-400. The authors state that, “The results of this study indicate that, to date, UP has been used in populations with seven different medical conditions…”   Actually, the list is much larger, and includes multiple sclerosis which not in the authors’ list.
  6. In sum, this study does not add any value to the existing UP protocol or expand our understanding of the transdiagnostic treatment of emotional disorders in medical conditions

References

 https://onlinelibrary.wiley.com/doi/full/10.1002/wps.20631

https://training.cochrane.org/handbook

https://www.equator-network.org/reporting-guidelines/prisma/

https://www.equator-network.org/reporting-guidelines/prisma-protocols/

https://www.crd.york.ac.uk/prospero/

 https://www.ncbi.nlm.nih.gov/pmc/articles/PMC6502428/

Author Response

  1. This systematic review attempts to collate all empirical evidence from the 9 studies that fit its specified eligibility criteria. This reviewer thinks the number of studies in this systematic review is too few to address a systematic review research objective.  

Response: Thank you for your comment. We agree with the reviewer that the number of studies is low. The UP is a recent treatment, which has already demonstrated its efficacy in the treatment of emotional disorders. Given that comorbidity between these and medical conditions is frequent, we considered it appropriate to carry out a review study to state the work that has been done so far and to identify gaps that research should address in future studies, thus stimulating research in this field. Nevertheless, we agree with the reviewer's comment and have added as a limitation of the study the few numbers of studies included.

Line 397-399: “Finally, some limitations should be taken into account when interpreting the results of this systematic review. First, a limited number of studies have been reported. In addition, most research studies were pilot or case studies as only four were RCTs, but they were the most rigorous and internally valid.”

  1. The nine studies include RCT (four studies), PILOT Studies (two), a single-case experimental study (two) and a case study(one), three study protocols, and the authors end mixing up apples and oranges in their analysis and synthesis. Case studies can be useful in research but they are at the very bottom of an evidence-based conclusion and pilot studies are typically used to gather evidence to refine a planned study design, not to use the study findings as evidence basis. Study protocols do not constitute an evidence base. Thus if we eliminate the case studies, study protocols, and pilot studies, the number of studies is even fewer.

 Response: We understand the concerns of the reviewer. A systematic review attempts to collate all empirical evidence that fits pre-specified eligibility criteria to answer a specific research question. Systematic reviews are considered essential tools to summarize available scientific information and identifying areas for future research.

In an early phase of research in a specific field or theme, which is the case of the application of the UP in medical conditions, reviews should include as much research data as possible, in order to know the current research interest and findings and identify gaps for future research. This is the aim of the present study, the first conducted in this specific theme. We understand that in subsequent phases, when an increase number of studies have been published, it should be appropriate to conduct a systematic review including only RCTs, or a meta-analysis study, or a systematic review of systematic reviews.

We totally agree with the reviewer that study protocols do not constitute an evidence base, and that is the reason why we present them separately from research studies (Table 5). The purpose of including study protocols in the review is to state future directions in terms of the UP application in this area.

  1. The systematic review should look into all the published articles across many years within the year limit of publication set in the eligibility criteria. What was the year limit of publication set in the eligibility criteria?

Response: We totally agree with the reviewer. Thank you very much for the question. There was no year limit of publication. We copy the sentence from the manuscript in which it is specified:

Line 120: “There were no language or publication period restrictions.”

  1. Page 12, lines 398-400. The authors state that, “The results of this study indicate that, to date, UP has been used in populations with seven different medical conditions…” Actually, the list is much larger, and includes multiple sclerosis which not in the authors’ list.

Response: Thanks for your comment. This sentence refers to the research studies included in the review (Table 1), of which two are conducted with population with multiple sclerosis. As mentioned in the limitations, “some factors may have biased the present systematic review findings, including the fact that only four databases were used for searches, and given the possibility of some studies offering contradictory results to those herein presented, which could change our conclusions” (Lines 401-404).

  1. In sum, this study does not add any value to the existing UP protocol or expand our understanding of the transdiagnostic treatment of emotional disorders in medical conditions

Response: From our point of view, this article is a reasonable contribution to show the existing knowledge regarding the use of the UP for the transdiagnostic treatment of emotional symptomatology or EDs in people with a medical condition diagnosed, and to identify gaps that research should address in future studies in this area.

Reviewer 2 Report

Thank you for giving me the opportunity to review the article. The author conducted a systematic review focusing on the unified protocol for the transdiagnostic treatment of emotional disorders. The topic is clinically important, but the manuscript may not be checked well before submission. Therefore, the reviewer thought that the manuscript should be revised carefully before further considerations. I only left the comments to complete the manuscript for submission. After the authors submit the corrected version, the reviewer comments about the contents.

Comments:

  1. The authors should check the author contributions section. There are many “XX”.
  2. In the main text of this manuscript, there are lots of “blind note”. The information is very important, and the authors should fill out these points before submitting the manuscript.
  3. The authors should mention about the names of the quality assessment methods in the Tables.

Author Response

  1. The authors should check the author contributions section. There are many “XX”.

Response: Thank you for this comment. We have added the initials in the authors contributions section (Lines 419-421).

  1. In the main text of this manuscript, there are lots of “blind note”. The information is very important, and the authors should fill out these points before submitting the manuscript.

Response: We totally agree with the reviewer. We have filled out this points as suggested (Lines 109, 136, 138, 140, 157, 158 and 161).

  1. The authors should mention about the names of the quality assessment methods in the Tables.

Response: Thank you very much for your suggestion. We have modified the title in the Tables to mention The Study Quality Assessment Tools from the National Heart Lung and Blood Institute (NHLBI) (Lines 385, 387, 389).

Reviewer 3 Report

Thank you for the opportunity to review this manuscript. Please find my review report attached.

Author Response

I have minor suggestions for improvement of the paper.

1. It is not clear if the inclusion criteria also consider prevention studies, although it is mentioned in the abstract and one study is in fact for prevention of depression. However, it was not clear for me since the aim is to know the “current research interest in using UP to treat emotional symptoms or EDs in people also with a medical condition”, one of the exclusion criteria is “did not test a treatment” and prevention purposes were not discussed elsewhere.

Response: We totally agree that the information should be better explained in the text. We have added this in the text and we have also included an additional comment  regarding this issue, meaning the need to carry out more prevention studies.

Line 101: “UP to prevent and treat emotional symptoms or EDs in people also with a medical condition”

Lines165: “research studies included in the systematic review, only one was a prevention study [43], while the rest were treatment studies.”

Line 332-333: “in using UP for preventing and treating emotional symptoms or EDs in people with a”

Line 362: “Another aspect worth mentioning is that treatment studies have drawn more interest than prevention studies.”

Line 416: “In addition, more prevention studies are needed.”

2. Also concerning the description of inclusion criteria according to PICOS, comparators and outcomes are not clearly stated.

Response: Thank you for highlight this point. We have rewritten the inclusion criteria section in order to clarify comparators and outcomes according to PICOS.

Lines122-125: “The present review included studies that met the following criteria: (1) were scientific articles (including case studies, pilot studies, randomized controlled studies, etc.) or a study protocol; (2) reported using UP or an adaptation of it; (3) included all comparators and outcomes; (4) included samples of all ages with a medical condition and emotional symptomatology or an ED diagnosis.”

3. Finally, throughout the manuscript, authors refer to “nine included studies”, what may be misleading, since 12 papers were included. I propose to use the terms ‘clinical trials’ and ‘study protocols’ (see lines 169-170), or using other terms that distinguish them in a more clear way.

Response: Thank you very much for this suggestion. In order to distinguish both types of articles in a clearer way, we have decided to use the terms “research study” and “study protocol”. We have made changes throughout the manuscript to change the terms. For example, in the abstract: “The aim of this systematic review is to know the current (research studies) and future research interest (study protocols) in using the UP for…”.

Specific comments

Introduction

Line 26 – I suggest including a brief definition of communicable and non-communicable diseases, or distinctive features.

Response: Thank you very much for your suggestion. We have added a definition of communicable and non-communicable diseases.

Line 28-31: “The former concern diseases caused by the transmission of a specific infectious agent from an infected source (person, animal, etc.) to a susceptible host [1]. While the latter are the result of a combination of genetic, physiological, environmental and behavioral factors, and are usually of long duration [2].”

Line 49 – I suggest “The mental disorders that most often”

Response: Thank you for your suggestion. We have reformulated this phrase as proposed.

Line 51: “The mental disorders that most often present comorbidity with medical conditions are”

Line 49 – In “comorbidly”, do the authors mean comorbidity?

Response: Thank you very much for highlighting this mistake, we have corrected it.

Line 51: “The mental disorders that most often present comorbidity with medical conditions are”

Line 101- Suggestion to reformulate: “Considering the UP’s efficacy, a meta-analysis and a recent systematic review have shown that”

Response: Thank you for your suggestion. We have reformulated this paragraph as proposed.

Line 93: “Considering the UP’s efficacy, a meta-analysis and a recent systematic review have”

Line 106 – Correct “that are comorbid to”

Line 108 – Correct “a randomized controlled trial (RCT)”

Response: Thank you very much for highlighting these mistakes, we have corrected both of them:

Line 97: “or anxious and depressive symptoms that are comorbid to medical conditions”

Line 98: “By way of example, a randomized controlled trial (RCT) with irritable bowel syndrome patients reported a significant”

 Methods

Lines 147-153 – This is usually presented in Results, according to PRISMA guidelines.

Response: Thank you so much for highlighting this fact. As the reviewer suggested, we have moved the “Search and Screening” information to the beginning of the results section (Lines 148-162)

Lines 157-158 – Authors report 4 questions per inclusion criteria, but only reported 3 inclusion criteria. Clarify.

Response: We totally agree with the reviewer. It was a mistake, because when we reported these four questions, they concerned the exclusion criteria and not the inclusion criteria. We have corrected the mistake in the text.

Line 158-160: “Previously, four questions were determined, one per exclusion criteria: (1) Is it a scientific article? (2) Does it test a treatment? (3) Does it apply the UP?; (4) Does it apply the UP in a sample with a medical condition? The excluded publications were grouped according to all four exclusion reasons (see Figure 1).”

Lines 162-167 – It is confusing to have several inter-rater agreement, it is not clear if inter-rater agreement was 100% or not. Authors should clarify this paragraph.

Response: Thank you very much for your comment. We agree with the reviewer that to have several inter-rate agreement is confusing. The inter-rater  agreement in relation to the articles finally included in the review was 100%. Therefore, in order to clarify this paragraph we have decided to maintain only this information.

Line 161-162: “After the study eligibility assessment phase, inter-rater agreement was calculated (Co-hen’s kappa). A 100% agreement of (Cohen's Kappa = 1) was reached.”

Results

I suggest to include the search results in the beginning of this section (see comment above).

Response: Thank you so much for this suggestion. As commented above, we have moved the “Search and Screening” information to the beginning of the results section (Lines 148-162)

Line 193 – Authors should clarify that they are referring to the treatment studies. It should be clear that the authors describe first the clinical trials and only after the study protocols.

Response: Thank you very much for this suggestion. In order to clarify and distinguish from study protocols, we have changed the term to “Research study”.

Line 163: “3.2. Characteristics of the Included Research Studies”

Line 205 – I suggest removing “In the participants”.

Response: Thank you for this comment. As suggested, we have removed “in the participants”.

Line 174: “The most frequent EDs were anxiety disorders and depression.”

Line 265 – Correct “used the”

Line 352 – Correct “RCTs”

Response: Thank you for highlighting this mistakes, we have corrected all of them.

Line 228: “to adolescents with chronic pain [43], another used the ESTEEM-SC in an HIV population”

Line 295: “Regarding the study design included in protocols, two will be RCTs [51,53] and the”

Lines 422-424 – This sentence is not clear “Four of the nine included studies were categorized as risk of bias because the quality of three was rated as fair”. Did the authors mean moderate risk of bias or low risk of bias?

Response: Thank you for highlighting this error, we have included the missing word in the sentence, fair, according to the classification of the Study Quality Assessment Tools of the National Heart Lung and Blood Institute.

Line 355: “Four of the nine included research studies were categorized as fair risk of bias be-cause the quality of three was rated as "fair" [41,42,48]”

Line 446 – Replace ‘PU’ for ‘UP’

Response: Thank you for highlight this mistake. We have corrected it:

Line 375: “We found different modifications made to UP in the research studies”

Line 446-448. This sentence seems to be incomplete, please confirm.

Response: We totally agree with the reviewer. Thank you very much for highlighting this error. We have corrected the sentence:

Line 375-377: “We found different modifications made to UP in the research studies, where its ability to adapt to the specific characteristics of each medical condition and the needs of those people who suffer from them are noteworthy”

 Table 1

“setting” – To be coherent with the section ‘Data Collection’, I suggest using the same term – “format” or “setting”

Response: Thank you for your suggestion. We have changed the term to “Setting” in the “Data Collection” section.

Line 132: “medical condition, emotional disorders, setting, number and frequency of sessions, primary”

Replace ‘PU’ for ‘UP’

Abbreviations: DASS-4 should be DASS-42; the following abbreviations should be in the end of the table: SCED, CESD, ODIS, MEAC.

Response: Thank you very much for highlighting these mistakes, we have corrected them in Table 1.

Table 5 - The following abbreviations should be in the end of the table: EuroQol, CVD.

Response: Thank you for highlight these mistakes, we have added EuroQol in the end of the table and corrected “CVD” to “CVE” (CVE, Cardiovascular event).

Figure 1 – In ‘Included’ I suggest “clinical trials” instead of “treatment studies”, to be in accordance with the main text (line 153)

Response: Thank you very much for this suggestion. As mentioned in the response to previous comments, we have decided to use the terms “Research study” and “Study protocol”. Thus, as the reviewer suggested, we have changed “Treatment studies” to “Research studies” in Figure 1.

Reviewer 4 Report

This topic is very relevant for the improvement of clinical practice and it is necessary to increase our understanding. In this sense, the authors have done a great effort to review, analyse and synthesize information that clarifies some of the findings related to this topic.

The introduction, as well as the objective and methodology are adequate.

In my opinion, the results are well described, although the table could be simplified to facilitate reading, for example, unifying the information of the participants in the same column.

Both the discussion and the conclusions are interesting and provide applications aimed at research and clinical practice.

I really liked reviewing this article.

Author Response

In my opinion, the results are well described, although the table could be simplified to facilitate reading, for example, unifying the information of the participants in the same column.

Response: Thank you for your suggestion. We have simplified Tables 1 and 2 unifying the reference and country information in one column and the study design and setting in another.

Round 2

Reviewer 1 Report

I have reviewed and provided my comments to the first version of this manuscriptipt. 

Author Response

Thanks again for your valuable feedback in the first revision.

Reviewer 2 Report

Thank you for giving me the opportunity to review the article. The author corrected the manuscript according to the comments by me and the other reviewers mostly. The reviewer thought that additional minor changes should be added before accepting the manuscript. I left my comments below.

Comments:

Methods:

  1. The authors should provide a PRISMA checklist as a supplementary material.

Results:

  1. Why the number of the articles which identified in the Medline were larger than the Pubmed? The Medline is covered by the Pubmed, and it also coverts non-Medline articles.
  2. The publication year is also important information. The authors should add it in the tables.

Author Response

Thank you for your valuable comments.
